# Changes in the bacterial community colonizing extracted and non-extracted tannin-rich plants in the rumen of dromedary camels

**Alaa Emara Rabee**[1]*, **Taha Abd El Rahman**[2,3], **Mebarek Lamara**[4]

**1** Animal and Poultry Nutrition Department, Desert Research Center, Cairo, Egypt, **2** Genetic Engineering and Biotechnology Research Institute, University of Sadat City, Sadat City, Egypt, **3** Centre SEVE, Department of Biology, Université de Sherbrooke, Sherbrooke, Canada, **4** Forest Research Institute, University of Quebec in Abitibi-Temiscamingue, Rouyn-Noranda, Canada

* rabee_a_m@yahoo.com

**Data Availability Statement:** All the sequences were deposited to the sequence read archive (SRA) under the accession number: PRJNA751263.

## Abstract

Leguminous trees and saltbushes provide potential alternatives to conventional feeds to overcome feed deficiency in arid and semi-arid countries. However, these plants are rich in antinutritional factors that have adverse effects on rumen microbiota and the host- animal. Some rumen microbiota detoxifies plants' secondary metabolites; thus, understanding plant-microbe interaction in the rumen could improve the plants' utilization. This study investigated the bacterial colonization and degradation of non-extracted and extracted tanniniferous plants: *Atriplex halimus*, *Acacia saligna*, and *Leucaena leucocephala*, in the rumen of three fistulated camels at 6 and 12 hours. The results showed that these plants have high nutritional value and tannins contents. The rumen degradation and microbial diversity of plant-attached bacteria varied according to plant type and phenols' extraction. Atriplex and leucaena showed higher microbial diversity at 6 and 12h, respectively. Bacteroidetes and Firmicutes were the main bacterial phyla, and the main genera were *Prevotella*, *RC9_gut_group*, *Butyrivibrio* that overrepresented in non-extracted plants (P<0.05). *Fibrobacteres* and *Anaerovibrio* showed sensitivity to plant toxins and *Ruminococcus* attached to plants with lower tannins. Several bacterial genera in the camel rumen have the potential to resist antinutritional factors in fodder plants, which could be used to improve the performance of grazing animals.

## 1. Introduction

Poor nutrition contributes significantly to the low productivity of ruminant animals in arid regions [1], wherever animal diets are derived mainly from low-protein and high-fiber roughages [2, 3]. Grazing trees and shrubs are promising alternative solutions for overcoming feed deficiency due to high protein, high digestible fiber, and soluble sugars [1, 2, 4]. Browse plants in North Africa are dominated by acacia, leucaena, and atriplex [1–6]. However, most tropical

**Funding:** The authors received no specific funding for this work.

**Competing interests:** The authors have declared that no competing interests exist.

trees and shrubs contain high levels of antinutritional factors (AF), such as phenolic compounds and tannins that decline animal performance by reducing palatability and feed intake, and inhibiting digestive enzymes and rumen microorganisms [1, 7]. For example, the inclusion of acacia in the sheep diet at a level of 300 g / head/ day without adaptations caused death after 9–21 days [8].

However, certain ruminant animals are able to detoxify these substances through rumen adaptation or by the presence of tannin-binding proteins in the saliva [7]. Dromedary camels have evolved distinctive traits and physiological mechanisms to survive in harsh desert circumstances [9]. Camels are non-selective grazers and utilize poor-quality fodder plants, including thorny bushes, halophytes, and bitter or toxic plant species that are usually avoided by other herbivores due to the higher activity of cellulolytic microorganisms and longer retention time of ingested forage in the rumen [5]. Therefore, the camel is a preferred animal in the production system in desert regions.

The rumen harbors highly diversified microbial groups consisting of bacteria, archaea, viruses, protozoa, and fungi that interact synergistically to convert indigestible plant biomass into microbial proteins and volatile fatty acids; additionally, rumen microbiota degrades plant toxins into utilizable products for host-animal [1, 10]. Bacteria represent the largest microbial group in the rumen and play a critical role in rumen fermentation process [11]. Therefore, any modulation in the rumen microbiome affects the metabolism in the rumen and the efficiency of the host-animal [10]. The rapid expansion of next-generation sequencing techniques has enabled determining the changes in rumen microbiota under various conditions, which offers the possibility to improve rumen fermentation [12]. AF in fodder plants inhibit the growth and digestion enzymes of rumen microorganisms [1, 13], which affects rumen fermentation negatively. Unlikely, some rumen microorganisms can degrade and utilize antinutritional substances [13–15]. Subsequently, rumen bacteria contribute to the adaptability of ruminants to tannin-rich fodders [16]. For instance, the bacterial genus *Synergistes jonesii* can degrade the mimosine, the main phytotoxin in leucaena, which helped the ruminant animals in Australia to consume leucaena without showing ill effects when they were inoculated by rumen fluid of Indonesian and Hawaiian goats that are rich in *Synergistes jonesii* [13].

Several chemical approaches have been developed to deactivate AF in fodder plants and improve digestibility [7]. However, previous studies revealed that tannin extraction does not remove all tannins and does not affect all plants equally [17–19]. Thus, it is necessary to describe the nutritional value as well as rumen microbial colonization and degradation of extracted and non-extracted browsing plants to develop proper utilization strategies [20–22]. There is a lack of information regarding the diversity and composition of rumen bacteria colonizing tannin-rich fodder plants. Therefore, this study applied PCR-amplicon sequencing through Illumina Mi-Seq to assess the differences in the diversity and structure of plant-attached bacterial communities along with degradation of non-extracted and acetone-extracted acacia, leucaena, and atriplex at different incubation times (6 and 12 hours) in the rumen of camels.

## 2. Methods

### 2.1. Plant collection and phenols extraction

Three plants were used in this study, including *Atriplex halimus*, *Acacia saligna*, and *Leucaena leucocephala*. All the plants were grown in the farm of Maryout Research Station, Desert Research Center, Alexandria, Ministry of Agriculture and Land Reclamation, Egypt. The plants were grown in separate plots in the same season under the same farm conditions. The samples of edible parts, including leaves and wet stems, were collected to be used in the study.

Adequate representative samples (1–2 kg) for each plant species were collected from 8 to10 different trees or shrubs in four sub-plots (70 m x 70 m). These samples were collected between May to July 2019 according to the regulations and permissions of the Range Management Unit and Maryout Research Station, Desert Research center, Egypt. The samples were pooled and oven-dried at 50 ˚C and then ground to pass a 1 mm sieve. Phenolic compounds in the ground plant samples were extracted by ultrasonic-assisted extraction using 70% aqueous acetone (v/v) according to the protocol of Makkar [23]. Briefly, about 100 g of dried and ground plant material was taken to a 2 L beaker; then 1000 ml of 70% aqueous acetone (v/v) was added, and the beaker was suspended in an ultrasonic water bath and subjected to ultrasonic treatment for 30 min (3 x 10 min with 5 min break in between) at room temperature. Then the supernatant was used to determine total phenols and total tannins. The pellets were air-dried, then oven-dried at 50 ˚C, and kept for further analysis. Subsequently, the analyses were conducted in three extracted and three non-extracted plants as follows: non-extracted atriplex (AN), extracted atriplex (AE), non-extracted acacia (CN), extracted acacia (CE), non-extracted leucaena (LN), and extracted leucaena (LE).

## 2.2. Animals and feeding

Three fistulated she-camels (*Camelus dromedarius*) (6-years age; average body weight, 450 kg) were used in this study. The camels were kept indoors individually and were fed *ad libitum* on Egyptian clover hay and allowed free drinking water. This study was conducted under guidelines set by the Department of Animal and Poultry Production, Desert Research Center, Egypt. The project was approved by the Institutional Animal Care and Use Committee, Faculty of Veterinary Medicine, University of Sadat City, Egypt (Reference: VUSC00008). All methods were performed in compliance with the ARRIVE guidelines.

## 2.3. *In situ* rumen incubation

Dried plant material (3g) was weighed into heat-sealed nylon bags (10 × 20 cm; pore size = 50 μm). Twenty-four sealed bags, four per plant, were placed into the rumen of each fistulated camel before morning feeding. The bags were retrieved from each camel's rumen after 6 hrs and 12 hrs. Furthermore, one blank bag was used for every plant to allow the correction for dry matter infiltration of sample bags. Two bags were removed for every plant at every incubation period. After the removal from camel rumen, the first bag was rinsed with cold water until the water ran clear; then squeezed and dried at 60˚C for 48 h. Dried samples were cooled in the desiccator and weighed for dry matter (DM) estimation. Dry samples were analyzed to estimate the content of crude protein (CP) and neutral detergent fiber (NDF) to determine the disappearance of dry matter (DMD), neutral detergent fiber (NDFD), and crude protein (CPD) after 6 and 12 hours using the difference method. The remaining bag was rinsed with sterilized distilled water until it ran clear and transferred to a sterilized 50 ml tube. Then, it was immediately frozen at -80˚C for microbes' dissociation and DNA extraction to analyze the fiber-attached bacteria.

## 2.4. Microbial cells recovery, DNA extraction, PCR amplification, and sequencing

The microbial cells' dissociation from plant samples was conducted according to the protocol described by Pope et al. [24]. Briefly, the frozen incubated plant samples were thawed and suspended in a 15 mL dissociation solution (0.1% Tween 80, 1% methanol, and 1% tertiary butanol (vol/vol), pH 2). The mixture was vortexed for 1 min and centrifuged at 500 x g for 20s, and then the supernatant was collected in a sterile 50 ml tube. This step was repeated two more

times, and supernatants for each sample were collected and pooled. Microbial cell pellets were collected by centrifugation at 12,000 × g for 5 min and subjected to DNA extraction by i-geno-mic Stool DNA Extraction Mini Kit (iNtRON Biotechnology, Inc.) according to the manufacturer's instructions. DNA was eluted in 50 μL elution buffer, and DNA quality and quantity were verified through agarose gel electrophoresis and Nanodrop spectrophotometer (Thermo Fisher Scientific, Madison, Wisconsin, USA). The V4-V5 region of the bacterial 16S ribosomal DNA gene was amplified using primers 515F (5′-GTGYCAGCMGCCGCGGTAA-3′) and 926R (5′-CCGYCAATTYMTTTRAGTTT-3′) [25]. PCR amplification was conducted in a thermal cycler under the following conditions: 94°C for 3 min; 35 cycles of 94°C for 45 s, 50°C for 60 s, and 72°C for 90s; and 72°C for 10 min. PCR-products purification and preparation for sequencing were conducted according to the protocol described by Comeau et al. [26]. The amplicons were then sequenced using the Illumina Mi-Seq system in Integrated Microbiome Resource (Dalhousie University, Canada).

## 2.5. Chemical analysis

The contents DM, NDF, and acid detergent fiber (ADF) were determined in plant samples before and after incubation in the rumen. DM was measured by drying the residual material for 48 h at 60 °C. ADF and NDF were determined by the method of Van Soest et al. [27] without sodium sulfite using ANKOM Technology Method. CP was determined according to AOAC [28]. Total phenols (TP) and total tannins (TT) were determined according to the protocol of Makkar [23] using of Folin-Ciocalteu method.

## 2.6. Bioinformatics analysis

All the paired-end (PE) Illumina raw sequences were processed in R (version 3.5.2) using the DADA2 pipeline (version 1.11.3) as described by Callahan et al. [29]. First, quality checks were conducted, and clean reads were denoised, dereplicated, and filtered for chimeras to generate Amplicon Sequence Variants (ASVs). Taxonomic assignment of sequence variants was performed using a combination of the functions assignTaxonomy and assignSpecies and was compared using the SILVA reference database.

## 2.7. Statistical analysis

The relative abundances of bacteria groups were tested for normality using the Shapiro–Wilk test, and variables deemed non-normal were then arcsine transformed for statistical analysis. The effect of plant type (S), plant extraction (E), and S*E interactions on the differences in DMD, CPD, and NDFD, as well as microbial diversity and abundance of bacterial taxa at different sampling times were studied using mixed ANOVA. The post hoc Duncan test was carried out to determine the significant differences at $P < 0.05$, and the Bonferroni correction method was applied to correct P-value. The effect of incubation time on the relative abundance of rumen bacteria was examined using unpaired T test. Linear Discriminate Analysis (LDA) and Bray Curtis Permutational Multivariate Analysis of Variance (PERMANOVA) tests were performed using the values of DMD, CPD, NDFD, and the relative abundances of dominant bacteria phyla to show the differences in community structure and to compare the clustering of plant samples. The statistical analyses were performed using the SPSS (v. 20.0) [30] and PAST [31] software. All the sequences were deposited to the sequence read archive (SRA) under the accession number: PRJNA751263.

**Table 1. Chemical composition (%) on dry matter basis of experimental fodder shrubs.**

| Fodder shrub | CP | NDF | ADF | Total phenols | Total tannins |
|---|---|---|---|---|---|
| *Atriplex halimus* | 24.85 | 51.11 | 26.62 | 5.66 | 4.16 |
| *Acacia saligna* | 19.98 | 53.50 | 31.45 | 7.18 | 6.01 |
| *Leucaena leucocephala* | 23.15 | 41.08 | 32.53 | 7.08 | 3.69 |

CP = Crude protein; NDF = Neutral detergent fiber; ADF = Acid detergent fiber

## 3. Results

### 3.1. Chemical composition

The chemical compositions (%) on the DM basis of plants are shown in Table 1. The CP content was higher in atriplex and leucaena compared to acacia. NDF differed between plants, and acacia revealed the highest content, followed by atriplex and leucaena. ADF was higher in leucaena and acacia compared to atriplex. Acacia showed higher TP, followed by leucaena and atriplex. Furthermore, higher TT was observed in acacia, followed by atriplex, and leucaena, respectively (Table 1).

### 3.2. Effect of plant type and phenol extraction on *in situ* degradation

The results revealed that the disappearance of nutrients, DMD, CPD, and NDFD, were affected significantly by plant type (S) and phenols extraction (E) (Table 2) ($P < 0.05$). CN had lower DMD and CPD at all incubation times while LE and AE had higher DMD after 6 and 12 h, respectively; in addition, AE showed higher CPD. Moreover, AE showed greater NDFD; also, CN and LN revealed lower NDFD at 6 and 12 h, respectively (Table 2).

### 3.3. Diversity of bacterial communities at different incubation times

The Illumina sequencing of the V4 region on the 16S rDNA gene in 36 samples generated 891,236 high-quality sequence reads, with a mean of 24,756 sequence reads per sample. Alpha diversity metrics, including Chao1, Shannon, Inverse Simpson, and Observed ASVs, were used to evaluate the similarity in bacterial communities attached to the extracted and

**Table 2. Disappearance (%) of dry matter (DMD), crude protein (CPD), neutral detergent fiber (NDFD) at 6 and 12h of incubation for extracted and non-extracted atriplex (AN, AE), acacia (CN, CE), leucaena (LN, LE) in the rumen of camels.**

| Incubation (h) | Atriplex | | Acacia | | Leucaena | | SEM | P value | | |
|---|---|---|---|---|---|---|---|---|---|---|
| | AN | AE | CN | CE | LN | LE | | S | E | S x E |
| **% DM disappearance (DMD)** | | | | | | | | | | |
| 6 | 35.65 | 42.71 | 22.42 | 28.46 | 28.43 | 45.13 | 2.2 | 0.0001 | 0.0001 | 0.0001 |
| 12 | 46.58 | 51 | 25.42 | 30.67 | 35.38 | 45.76 | 2.4 | 0.0001 | 0.0001 | 0.0001 |
| **% CP disappearance (CPD)** | | | | | | | | | | |
| 6 | 38.46 | 49.78 | 8.57 | 16.72 | 12.22 | 19.30 | 2 | 0.0001 | 0.0001 | 0.0001 |
| 12 | 42.30 | 50.94 | 13.10 | 21.27 | 16.72 | 21.17 | 1.5 | 0.0001 | 0.0001 | 0.0001 |
| **% NDF disappearance (NDFD)** | | | | | | | | | | |
| 6 | 28.92 | 34.26 | 20.81 | 24.43 | 23.19 | 26.29 | 1 | 0.0001 | 0.0001 | 0.008 |
| 12 | 32.64 | 43.23 | 25.87 | 30.86 | 25.39 | 27.39 | 1.7 | 0.0001 | 0.0001 | 0.0001 |

SE = Standard Errors; AN = Non-extracted atriplex; AE = Extracted atriplex; CN = Non-extracted acacia; CE = Extracted acacia; LN = Non-extracted leucaena;
LE = Extracted leucaena.

**Table 3. Average of ASVs number, Chao1, Shannon, and Inverse Simpson indices of bacterial community colonizing to extracted and non-extracted atriplex, acacia, and leucaena at 6 and 12 hours incubations.**

| Incubation time (h) | Atriplex | | Acacia | | Leucaena | | SEM | P value | | |
|---|---|---|---|---|---|---|---|---|---|---|
| | AN | AE | CN | CE | LN | LE | | S | E | S x E |
| ASVs | | | | | | | | | | |
| 6 | 2822 | 2157 | 962 | 1741 | 1210 | 1296 | 242 | 0.02 | 0.9 | 0.6 |
| 12 | 1087 | 1390 | 1332 | 916 | 1950 | 1638 | 151 | 0.08 | 0.7 | 0.7 |
| Chao1 | | | | | | | | | | |
| 6 | 2831 | 2163 | 963 | 1745 | 1214 | 1300 | 242 | 0.02 | 0.9 | 0.6 |
| 12 | 1089 | 1394 | 1335 | 918 | 1958 | 1646 | 151 | 0.088 | 0.7 | 0.7 |
| Shannon | | | | | | | | | | |
| 6 | 7.3 | 7.06 | 6.4 | 6.8 | 6.6 | 6.75 | 0.1 | 0.055 | 0.75 | 0.58 |
| 12 | 6.5 | 6.7 | 6.6 | 6.4 | 7 | 6.9 | 0.1 | 0.27 | 0.93 | 0.77 |
| InvSimpson | | | | | | | | | | |
| 6 | 973 | 721 | 483 | 834 | 602 | 679 | 80 | 0.55 | 0.74 | 0.4 |
| 12 | 489 | 573 | 643 | 483 | 969 | 708 | 70 | 0.19 | 0.44 | 0.6 |

non-extracted plants (Table 3). Alpha diversity analysis showed a significant effect of plant type on ASVs number and Chao1 index at 6h, whenever CN showed lower diversity and AN had higher diversity (Table 3). In addition, incubation time did not reveal any effect on the diversity indices (S1 File). Beta diversity analysis was calculated and visualized using principal coordinate analysis based on the Bray–Curtis distances. The analysis revealed that bacterial communities were not distinctively separated by neither plant types nor phenol extraction (Fig 1).

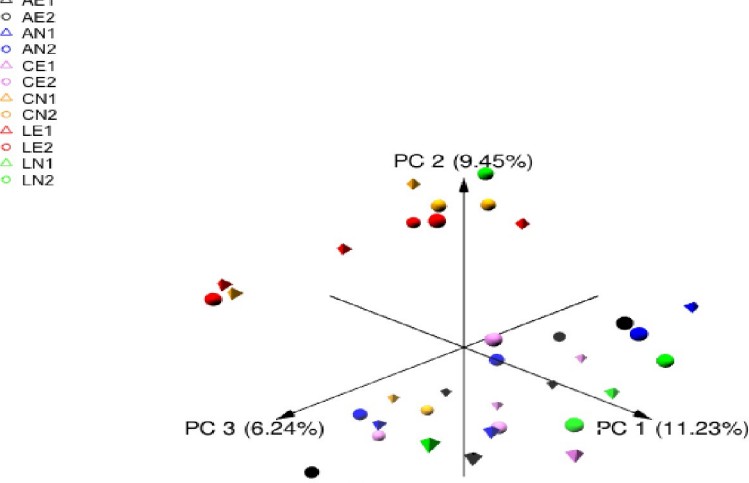

**Fig 1. Principal coordinate analysis; PCoA of attached bacterial community based on Bray-Curtis dissimilarity across extracted and non-extracted plants in the rumen of camel.** AE1 for extracted atriplex at 6h, AE2 for extracted atriplex at 12h, AN1 for non-extracted atriplex at 6h, AN2 for non-extracted atriplex at 12h. Also, CE1 for extracted acacia at 6h, CE2 for extracted acacia at 12h, CN1 for non-extracted acacia at 6h, CN2 for non-extracted acacia at 12h. Moreover, LE1 for extracted leucaena at 6h, LE2 for extracted leucaena at 12h, LN1 for non-extracted leucaena at 6h, LN2 for non-extracted leucaena at 12h.

### 3.4. Analysis of bacterial community

A total of 16 bacterial phyla were observed in this study, out of which six phyla were observed in all samples, including Bacteroidetes, Firmicutes, Planctomycetes, Proteobacteria, Spirochaetes, and Tenericutes. Plant type (S) and phenols extraction (E) affected the relative abundance of some bacterial phyla (Table 4). The phylum Bacteroidetes dominated the bacterial community and showed higher relative abundance in LE, and the lower relative abundance was observed with AN (Table 4) ($P<0.05$). The results revealed that AN showed higher proportions of Firmicutes, Proteobacteria, and Tenericutes ($P<0.05$). Furthermore, LE showed a lower relative abundance of Firmicutes, AE revealed a lower abundance of Proteobacteria, and CN retained less Tenericutes.

In this study, ten phyla were not observed in all samples, including Actinobacteria, Chloroflexi, Cyanobacteria, Fibrobacteria, Fusobacteria, Kiritimatiellaeota, Lentisphaerae, Patescibacteria, Synergistetes, Verrucomicrobia (S1 Table). Phylum Actinobacteria, Chloroflexi, Lentisphaerae, and Patescibacteria were found only in AN and CE. Phylum Cyanobacteria was disappeared from AE and CE, and Synergistetes was disappeared from LE. Fibrobacteria was associated only with extracted plants (AE, CE, and LE). Verrucomicrobia was observed in atriplex (AN and AE), acacia (CN and CE), and extracted leucaena (LE). Fusobacteria was observed only in AN and leucaena (LN and LE) (S1 Table). Moreover, the relative abundances of bacterial phyla were not changed due to increasing the incubation time form 6 h to 12 h except for phylum Bacteroidetes that was increased in group CE, and phylum Spirochaetesas that was increased in CN group as a result of increasing incubation time (S1 File).

The bacterial community attached to extracted and non-extracted plants was affiliated with 29 families. On the genus level, 78 genera were observed, out of which 56 genera were not shared in all samples (Table 5, S1 Table). Phylum Bacteroidetes was dominated by the family Prevotellaceae, Rikenellaceae, F082, p-251-o5, and S11_gut_group. Family Prevotellaceae was

**Table 4. The average of relative abundance (%) of dominant bacterial phyla colonized to extracted and non-extracted atriplex, acacia, and leucaena at 6 and 12 hours incubations.**

| Incubation time (h) | Atriplex | | Acacia | | Leucaena | | SEM | P value | | |
|---|---|---|---|---|---|---|---|---|---|---|
| | AN | AE | CN | CE | LN | LE | | S | E | S x E |
| **Bacteroidetes** | | | | | | | | | | |
| 6 | 66 | 86.4 | 84.4 | 83.7 | 84.7 | 87 | 2.35 | 0.01 | 0.002 | 0.01 |
| 12 | 68 | 87.6 | 84 | 88 | 86.6 | 88 | 3.11 | 0.41 | 0.22 | 0.51 |
| **Firmicutes** | | | | | | | | | | |
| 6 | 25.9 | 9.4 | 12 | 11.5 | 11.7 | 8.8 | 1.8 | 0.006 | 0.014 | 0.016 |
| 12 | 27.1 | 8.3 | 12.4 | 7.5 | 8.9 | 8.1 | 1.7 | 0.41 | 0.23 | 0.5 |
| **Planctomycetes** | | | | | | | | | | |
| 6 | 0.8 | 1 | 0.6 | 1 | 0.7 | 0.7 | 0.13 | 0.936 | 0.27 | 0.58 |
| 12 | 0.8 | 0.3 | 0.1 | 0.4 | 0.4 | 0.3 | 0.09 | 0.56 | 0.78 | 0.35 |
| **Proteobacteria** | | | | | | | | | | |
| 6 | 4 | 0.17 | 1.4 | 0.4 | 0.5 | 0.4 | 1 | 0.004 | 0.001 | 0.003 |
| 12 | 1.45 | 0.94 | 1.2 | 1.2 | 0.2 | 0.2 | 0.8 | 0.048 | 0.7 | 0.8 |
| **Spirochaetes** | | | | | | | | | | |
| 6 | 1.3 | 0.8 | 0.7 | 0.9 | 0.3 | 0.5 | 0.19 | 0.57 | 0.97 | 0.6 |
| 12 | 0.8 | 1.4 | 0.95 | 0.4 | 1.4 | 1.1 | 0.19 | 0.58 | 0.44 | 0.5 |
| **Tenericutes** | | | | | | | | | | |
| 6 | 1.98 | 1.8 | 0.2 | 1.6 | 0.3 | 0.8 | 0.2 | 0.008 | 0.14 | 0.28 |
| 12 | 1.98 | 1.06 | 0.33 | 1.05 | 1.2 | 0.7 | 0.1 | 0.2 | 0.95 | 0.48 |

**Table 5. The of relative abundance (%) of dominant bacterial families and genera colonizing extracted and nonextracted plants at 6 and 12 hours.**

| | Incubation time (h) | Atriplex | | Acacia | | Leucaena | | SEM | P value | | |
|---|---|---|---|---|---|---|---|---|---|---|---|
| | | AN | AE | CN | CE | LN | LE | | S | E | S x E |
| P: Bacteroidetes; C: Bacteroidia; O: Bacteroidales; F: Prevotellaceae | | | | | | | | | | | |
| Prevotellaceae, F | 6 | 44.3 | 64.1 | 45.3 | 39.2 | 52.5 | 54.6 | 2.2 | 0.008 | 0.95 | 0.009 |
| | 12 | 44.8 | 44.3 | 42.1 | 40.1 | 53.3 | 64.2 | 2 | 0.278 | 0.55 | 0.48 |
| Prevotella_1,G | 6 | 35.9 | 56.4 | 31.5 | 25.9 | 38.3 | 41 | 2 | 0.0001 | 0.36 | 0.037 |
| | 12 | 31.3 | 34.2 | 31.5 | 27.2 | 40 | 54 | 1.9 | 0.0001 | 0.74 | 0.01 |
| Prevotella_7, G | 6 | 1.2 | 1.4 | 2.3 | 0.5 | 0.8 | 0.4 | 0.25 | 0.48 | 0.1 | 0.01 |
| | 12 | 5.2 | 1.2 | 2.4 | 0.8 | 1.4 | 2.1 | 0.4 | 0.8 | 0.28 | 0.43 |
| P: Bacteroidetes; C: Bacteroidia; O: Bacteroidales; F: Rikenellaceae | | | | | | | | | | | |
| Rikenellaceae, F | 6 | 8.3 | 6.7 | 8.5 | 6.3 | 6 | 4.7 | 0.3 | 0.17 | 0.1 | 0.9 |
| | 12 | 7.6 | 6.5 | 8 | 7 | 7.9 | 6.5 | 0.29 | 0.84 | 0.19 | 0.96 |
| RC9_gut_group,G | 6 | 8 | 6.4 | 8.3 | 6 | 5.9 | 4.5 | 0.3 | 0.159 | 0.09 | 0.9 |
| | 12 | 7.4 | 6.3 | 8 | 6.9 | 7.7 | 6.3 | 0.25 | 0.8 | 0.2 | 0.9 |
| NA (Non- classified),G | 6 | 6.6 | 7.6 | 6.6 | 21.1 | 13.2 | 12.8 | 1.3 | 0.31 | 0.02 | 0.02 |
| | 12 | 7.6 | 17 | 10.7 | 12.7 | 9.2 | 7.35 | 3 | 0.8 | 0.2 | 0.2 |
| P: Bacteroidetes; C: Bacteroidia; O: Bacteroidales | | | | | | | | | | | |
| F: F082; NA, G | 6 | 3.8 | 5.6 | 7.8 | 6.2 | 5.8 | 4.7 | 0.4 | 0.28 | 0.75 | 0.33 |
| | 12 | 5.1 | 5.7 | 5.3 | 6.2 | 7.3 | 5.8 | 0.35 | 0.78 | 0.9 | 0.8 |
| F: p-251-o5; NA, G | 6 | 2 | 1.4 | 13.9 | 9.2 | 6.2 | 8.9 | 1.3 | 0.017 | 0.6 | 0.2 |
| | 12 | 2 | 12.8 | 16.1 | 21.15 | 7.4 | 3 | 1 | 0.049 | 0.28 | 0.21 |
| F: S11_gut_group; NA, G | 6 | 0 | 0.12 | 0.4 | 0.29 | 0.11 | 0.1 | ND | 0.18 | 0.17 | 0.38 |
| | 12 | 0.14 | 0.25 | 0.27 | 0.25 | 0.28 | 0.26 | 0.02 | 0.2 | 0.19 | 0.42 |
| P: Firmicutes; C: Clostridia; O: Clostridiales: F: Lachnospiraceae | | | | | | | | | | | |
| F: Lachnospiraceae | 6 | 10.9 | 2.8 | 2.7 | 3.3 | 4.3 | 3.8 | 1.2 | 0.097 | 0.036 | 0.022 |
| | 12 | 17.8 | 3.2 | 3.06 | 2.3 | 3.7 | 3.3 | 1.7 | 0.3 | 0.26 | 0.36 |
| Butyrivibrio_2, G | 6 | 1.1 | 0.2 | 0.6 | 0.55 | 0.5 | 0.3 | 0.07 | 0.03 | 0.0001 | 0.005 |
| | 12 | 0.9 | 0.31 | 1.1 | 0.46 | 0.7 | 0.23 | 0.05 | 0.001 | 0.0001 | 0.02 |
| F: Ruminococcaceae | 6 | 7.3 | 4.7 | 3.8 | 6.6 | 3.8 | 3.7 | 0.4 | 0.014 | 0.034 | 0.014 |
| | 12 | 3 | 3.9 | 3.1 | 4.2 | 3.6 | 3.4 | 0.3 | 0.9 | 0.3 | 0.5 |
| P: Firmicutes; C: Negativicutes; O: Selenomonadales: F: Veillonellaceae | | | | | | | | | | | |
| Veillonellaceae, F | 6 | 5.4 | 0.93 | 4.7 | 0.66 | 3.1 | 0.9 | 0.6 | 0.03 | 0.0001 | 0.05 |
| | 12 | 5 | 0.6 | 5.8 | 0.48 | 1.09 | 1 | 0.65 | 0.004 | 0.02 | 0.02 |
| P: Firmicutes; C: Clostridia; O: Clostridiales: F: Family_XIII | | | | | | | | | | | |
| Family_XIII, F | 6 | 0.71 | 0.33 | 0.27 | 0.37 | 0.34 | 0.27 | 0.04 | 0.02 | 0.13 | 0.09 |
| | 12 | 0.3 | 0.36 | 0.14 | 0.3 | 0.3 | 0.1 | 0.035 | 0.09 | 0.54 | 0.06 |
| Anaerovorax, G | 6 | 0.45 | 0.23 | 0.24 | 0.14 | 0.23 | 0.13 | 0.02 | 0.003 | 0.02 | 0.18 |
| | 12 | 0.27 | 0.21 | 0.11 | 0.3 | 0.11 | 0.13 | 0.015 | 0.159 | 0.16 | 0.055 |
| P: Planctomycetes; C: Planctomycetacia;O: Pirellulales: F: Pirellulaceae | | | | | | | | | | | |
| p-1088-a5_gut_group, G | 6 | 0.5 | 0.75 | 0.42 | 1 | 0.4 | 0.67 | 0.07 | 0.95 | 0.053 | 0.53 |
| | 12 | 0.4 | 0.26 | 0.07 | 0.4 | 0.33 | 0.4 | 0.06 | 0.71 | 0.68 | 0.21 |
| P: Spirochaetes; C: Spirochaetia; O: Spirochaetales: F: Spirochaetaceae | | | | | | | | | | | |
| Treponema_2, G | 6 | 1.2 | 0.7 | 0.6 | 0.75 | 0.4 | 0.67 | 0.12 | 0.78 | 0.94 | 0.57 |
| | 12 | 1.03 | 1.2 | 0.86 | 0.26 | 1.3 | 0.8 | 0.1 | 0.58 | 0.33 | 0.43 |

P = Phylum, C = class, O = order, F = family, G = genera

dominated by the genus *Prevotella*, which was higher in AE compared to other groups (P<0.05). Family Rikenellaceae was dominated by *RC9_gut_group* and Non-classified genus that were overrepresented in non-extracted plants (AN, CN, and LN) (Table 5). Additionally, the genus *Alloprevotella* was not shared in all samples and was observed in non-extracted plants (S1 Table).

Phylum Firmicutes was dominated by four families, Lachnospiraceae, Ruminococcaceae, Veillonellaceae, and Family_XIII, that were affected by plant type (Table 5, S1 Table). Genus *Butyrivibrio* dominated family Lachnospiraceae and was influenced by plant type and tannin extraction and was higher in non-extracted plants (AN and CN) (P<0.05). Genus *Oribacterium* and *Shuttleworthia*, within the family Lachnospiraceae, were not shared in all samples and were observed in AN, CN, and LN. Family Ruminococcaceae was higher in AN than other plants (P<0.05). Other genera that were not shared in all samples within phylum Firmicutes, including *Selenomonas* and *Anaerovibrio* in family Veillonellaceae; besides *Lactobacillus* in family Lactobacillaceae. These genera were observed in some non-extracted plants (S1 Table).

Members of the phylum Spirochaetes were assigned mainly to the genus *Treponema* (Table 5). Phylum Synergistetes was classified into two genera, *Fretibacterium* and *Pyramidobacter* (S1 Table). *Pyramidobacter* was observed in non-extracted plants, and *Fretibacterium* disappeared from LE. Phylum Proteobacteria was dominated by the genus *Desulfovibrio*. Phylum Planctomycetes was dominated by the family Pirellulaceae and genus p-1088-a5_gut_group (S1 Table). Additionally, few bacterial groups were affected by increasing incubation time, including family Ruminococcaceae that was declined in group CE and genus *Treponema* that was increased in group LN by increasing incubation time (S1 File).

**3.4.1. Linear discriminant analysis (LDA) and Bray Curtis Permutational Multivariate Analysis of Variance (PERMANOVA).** LDA and PERMANOVA were conducted based on DMD, CPD, and NDFD and the relative abundances of dominant phyla (Fig 2a and 2b). The results of LDA revealed that the libraries were separated distinctly by plant type. The differences between samples were driven by DMD, CPD, and NDFD and the relative abundance of phylum Bacteroidetes and Firmicutes, these findings were supported by a significant difference (P = 0.0001) obtained by PERMANOVA test at 6h and 12h. This result indicated clear variation in rumen degradation and plant-attached microbial communities, which confirms that the chemical composition of fodder plants affected the rumen microbiota and ruminal degradation.

# 4. Discussion

Antinutrition factors in fodder plants have an adverse effect on rumen microorganisms and host- animals [32]. Therefore, understanding the interaction between rumen microorganisms and anti-nutritional-rich plants is the bottleneck to several implications, including: (1) improving strategies for utilizing grazing plants to fill the gap in animal feeding [1, 33], and (2) transfer the tannin-resistant microbial isolates or consortiums to susceptible animals to improve their grazing efficiency [13, 16]. In this study, the colonization of non-extracted and extracted atriplex, acacia, and leucaena by rumen bacteria in camels was studied at different incubation times (6 and 12h). Forage plants in this study had different CP, NDF, ADF, TP, and TT contents (Table 1). Chemical compositions of plant tissues vary according to many factors, including plant species, growing season, geographical location, growth stage, plant part, and tannin-extraction method [7, 23, 32, 34]. Atriplex showed higher CP and NDF contents than acacia and leucaena (Table 1), which was also indicated by Sallam et al. [34]. The percentages of plants' CP, NDF, and ADF were in the ranges shown by similar studies [3, 4, 34, 35]. Moreover, acacia and leucaena had higher total phenols compared to atriplex, while acacia

a- 6h

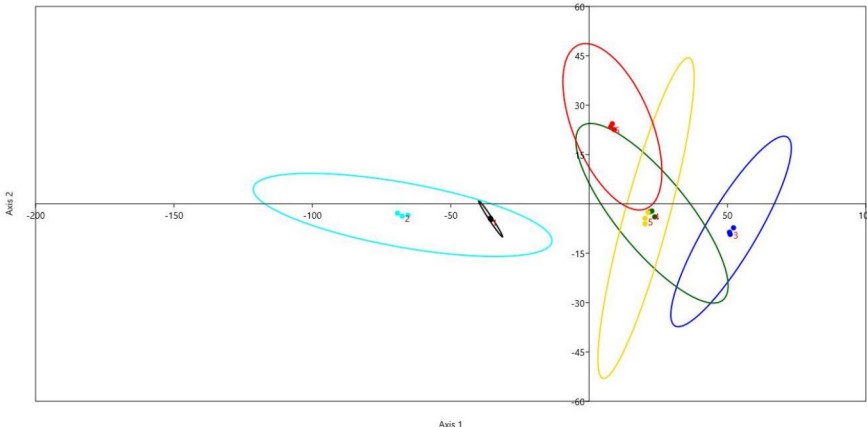

b- 12h

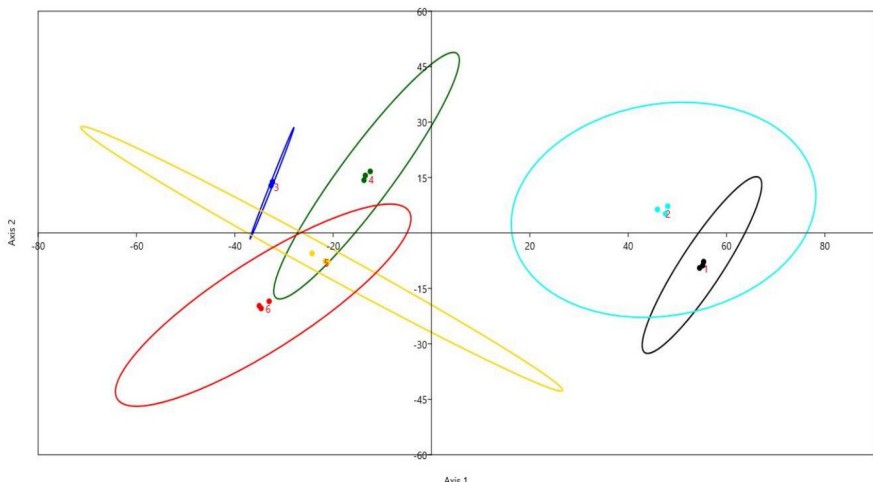

**Fig 2.** Linear Discriminate Analysis; LDA based on the values of disappearance of dry matter (DMD), neutral detergent fiber (NDFD), and crude protein (CPD) as well as the relative abundances of dominate bacterial phyla and genera attached to extracted and non-extracted plants in the rumen of camel at 6h (A) and 12 h (B); non-extracted atriplex (black dots), extracted atriplex (aqua dots), non-extracted acacia (blue dots), extracted acacia (dark green dots), non-extracted leucaena (gold dots), and extracted leucaena (red dots).

showed the highest tannins content (Table 1). Similar findings were obtained by other investigators [3, 4, 34]. However, the CP and TT content of atriplex and acacia were higher than the results of Khattab [32], which could be attributed to using different plant parts or the difference in the growth season.

Besides phenolic compounds and tannins, forage plants have specific types of AF [1]. Leucaena is known to have a non-protein amino acid mimosine that is toxic to animals [13]. Atriplex contains different secondary metabolites, such as tannins, flavonoids, saponins, oxalates, alkaloids, and resins [6, 36]. Several antinutritional compounds were detected in acacia, including tannins, oxalate, saponins, cyanogenic glycoside, and fluoroacetate [1, 15]. Phenolic compounds have different molecular weights and chemical structures [17]. Thus, the response of plant species to specific extraction methods could be varied [19]. Additionally, removing

tannins from fodder plants is accompanying the loss of plant nutrients such as proteins, minerals, soluble sugars, and polysaccharides [7, 18]. Accordingly, the degradation and bacterial colonization of extracted and non-extracted plants in the rumen will be affected [33]. The disappearance of nutrients (DMD, CPD, and NDFD) varied according to plant type and plant extraction, which is supported by the results of LDA (Fig 2; Table 2) and agrees with previous studies [4, 37]. Lower degradation of CN could be attributed to higher NDF, phenols, and tannins, which confirm that the chemical composition of fodder shrubs affected the rumen microbiota and the ruminal degradation [33, 34]. On the other hand, higher degradation in the atriplex could be attributed to higher content of non-protein nitrogen compounds [38]. Previous studies found that the rumen disappearance of the diet's nutrients was declined by the inclusion of acacia, leucaena, and atriplex in the animal diet [3, 39].

The lowest alpha diversity in CN indicates low bacterial colonization and could be attributed to higher TP, TT, and NDF, which is in agreement with other studies [8, 22, 33, 39]. Most bacteria attached to plants were affiliated with the phylum Bacteroidetes and Firmicutes (Table 4). This result is similar to those obtained in other studies on hay and straw [22], lignocellulosic forages [40], and *Lotus corniculatus*, which have high-tannin content [33]. On the other hand, this finding agrees with studies on solid-attached microorganisms in camels [41]. To the best of our knowledge, this study is the first to investigate the microbial colonization of extracted and non-extracted tannin-rich fodder plants in the rumen of dromedary camel.

The phylum Bacteroidetes predominated the bacterial community in all samples. This phylum degrades a wide range of substrates, including cellulose, pectin, and soluble polysaccharides. Moreover, unclassified Bacteroidetes are more specialized in lignocellulose degradation [42], which highlights the prevalence of this phylum in extracted acacia and leucaena (CN and LN) that have higher ADF. Most Bacteroidetes members were assigned to the genus *Prevotella* (Table 5), which uses different substrates in the growth, including hemicellulose, pectin, proteins, and peptides [22, 41]. This genus was declined in some extracted plants, which could be explained by the depletion of growth substrates and highlights its resistance to plant tannins [37]. In addition, a positive association between protein content and *Prevotella* was reported [22], which illustrates the higher representation of this genus in atriplex and leucaena that have higher CP compared with acacia. Genus *Alloprevotella* was reported to be adapted to condensed tannin, which might illustrate the prevalence of this genus in non-extracted plants (S1 Table) [43, 44]. Uncultured Rikenellaceae (Phylum Bacteroidetes), such as *RC9_gut_group*, was higher in non-extracted plants, indicating the ability of this group to withstand plant-tannins (Table 5) [43].

Genus *Butyrivibrio* and *Ruminococcus* dominated phylum Firmicutes and showed higher abundance with atriplex and leucaena, with higher NDF (Table 5). These genera are polysaccharides-degrading bacteria, and *Butyrivibrio* has proteolytic activities [22, 40]. Also, *Butyrivibrio* degrades mimosine and tannin, explaining the higher abundance of this genus of non-extracted leucaena (LN) that has mimosine [7, 13]. Some species of *Ruminococcus* showed higher relative abundance in non-extracted atriplex, and other species increased in acacia after the extraction (S1 Table), which is in line with previous studies [43, 45] that indicated that this genus resists specific types of tannins or polyphenols. Bacterial genera within phylum Firmicutes that were not found in all samples, including *Oribacterium* and *Shuttleworthia* (family Lachnospiraceae), *Selenomonas* (family Veillonellaceae), *Lactobacillus* (family Lactobacillaceae) (S1 Table); these genera were found in non-extracted plants, which highlight their adaptability to phenolic compounds [16, 46, 47]. Genus *Treponema* dominated the phylum Spirochaetes showed higher adaptability to different phenolic compounds [44].

*Fibrobacteria*, the main cellulolytic bacteria in the rumen [48], is highly sensitive to phenolic compounds and tannins [49], which illustrates the presence of the phylum only in extracted

plants (S1 Table). In our study, the phylum Synergistetes was dominated by the genera *Fretibacterium* and *Pyramidobacter* (S1 Table), which have the adaptability to phenols compounds, especially thymol and fluoroacetate, that are toxic compounds found in acacia [50, 51]. In addition, the genus *Pyramidobacter* has a potential role in cellulose degradation, highlighting the presence of Synergistetes phylum in non-extracted acacia and atriplex and the increase in the relative abundance after extraction [15, 52, 53]. Some of the bacterial genera in this study were previously reported as resistant isolates to alkaloids, such as *Anaerovibrio* (phylum: Firmicutes), *Desulfovibrio* (phylum: Proteobacteria), and *Prevotella* (phylum: Bacteroidetes) [15], which explain their prevalence in non-extracted plants (S1 Table). Also, the genus *Anaerovibrio* showed adaptability to tannin from chestnut extract [44], which supports the prevalence of this genus in non-extracted atriplex. *Anaerovibrio* has an essential role in lipid metabolism and biohydrogenation in the rumen [54]. Our findings showed that this genus was not found in acacia and leucaena, which agrees with previous results [55, 56] indicating that polyphenols adversely affect this genus. Consequently, acacia and leucaena extracts could be used to reduce the biohydrogenation of unsaturated fatty acids to improve the quality of milk and meat [57].

Our results highlight the atriplex and leucaena as appropriate fodder plants in animal feeding, which agrees with previous studies [3, 6, 35]. In addition, using acacia in animal feeding requires different strategies to reduce the negative effect of antinutritional factors, such as adapted feeding, chemical treatments, and transferring rumen content of adapted animals to other unadapted animals [1, 7, 8, 13].

Animal species and the chemical composition of animal diets are the main determiners of the rumen microbiome composition [41]; our results show that rumen bacteria in camels could be resistant to plant toxic compounds. In addition, previous studies of the rumen microbiome of camel and other ruminant animals [41, 58, 59], alongside with current study, showed that camel rumen harbors greater density and diversity of fibrolytic or potential fibrolytic bacteria (S2 Table). For example, phylum Fibrobacteres represented 4.4% of rumen bacteria in camel [41]; while it accounted for 0.18% in sheep [58]; and 0.01% and 0.29% in cow and buffalo, respectively [59].

Furthermore, the relative abundance of rumen microbiota varies among forage types due to the differences in chemical compositions, which influence the colonization of rumen microbiota [22, 40]. Rumen bacteria of fistulated camels under investigation were previously investigated by Rabee et al. [41] whenever animals were fed Egyptian clover hay; and the comparison between the results of that study and the current study revealed that the relative abundances of main bacterial genera were affected by the forage type (S2 Table). For example, the family Ruminococcaceae represented 3 to 7.3% in the current study and 15.66% in the previous study [41].

It is essential to study secondary metabolites in fodder plants and the interaction between plants and gut microbiota. These compounds have a range of biological and medical activities as antimicrobial, anti-inflammatory, hepatoprotective, antithrombotic, anticarcinogenic, cardioprotective, and vasodilatory effects [36]. Furthermore, plant extracts could be used as safe alternatives to antibiotics and stimulate the growth of gut microflora and metabolism in the intestine [60].

## 5. Conclusions

The chemical composition of fodder plants and their contents of antinutritional factors are the main determiner of microbial colonization and degradation in the rumen. Plant extracts from acacia and lucaena could be used as a modifier to the rumen ecosystem. This work

demonstrates the adaptability of camels to plant toxins; therefore, rumen content from camels could be a source of detoxifier bacteria to be inoculated to unadapted animals.

## Supporting information

**S1 Table. The relative abundance (%) of rare bacterial phyla and bacterial families and genera colonized to extracted and non-extracted atriplex, acacia, and leucaena at 6 and 12 hours incubations.**
(DOCX)

**S2 Table. Comparison of the relative abundances (%) of the main rumen bacteria in camels of current study and previous study and other ruminant animals.**
(DOCX)

**S1 File. Effect of incubation time on the microbial diversity and the relative abundances of bacterial phyla and genera.**
(XLSX)

## Author Contributions

**Conceptualization:** Alaa Emara Rabee, Taha Abd El Rahman, Mebarek Lamara.

**Data curation:** Alaa Emara Rabee, Taha Abd El Rahman, Mebarek Lamara.

**Formal analysis:** Alaa Emara Rabee, Taha Abd El Rahman, Mebarek Lamara.

**Funding acquisition:** Alaa Emara Rabee.

**Investigation:** Alaa Emara Rabee, Taha Abd El Rahman, Mebarek Lamara.

**Methodology:** Alaa Emara Rabee, Taha Abd El Rahman, Mebarek Lamara.

**Project administration:** Alaa Emara Rabee.

**Resources:** Alaa Emara Rabee, Mebarek Lamara.

**Software:** Alaa Emara Rabee, Mebarek Lamara.

**Supervision:** Alaa Emara Rabee.

**Validation:** Alaa Emara Rabee, Taha Abd El Rahman, Mebarek Lamara.

**Visualization:** Alaa Emara Rabee, Taha Abd El Rahman, Mebarek Lamara.

**Writing – original draft:** Alaa Emara Rabee, Taha Abd El Rahman, Mebarek Lamara.

**Writing – review & editing:** Alaa Emara Rabee, Taha Abd El Rahman, Mebarek Lamara.

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
