## [Decision Letter · Decision Letter 0]

4 Jan 2023

PONE-D-22-27641Changes in the bacterial community colonizing extracted and non-extracted tannin-rich plants in the rumen of dromedary camelsPLOS ONE

Dear Dr. Rabee,

Thank you for submitting your manuscript to PLOS ONE. After careful consideration, we feel that it has merit but does not fully meet PLOS ONE’s publication criteria as it currently stands. Therefore, we invite you to submit a revised version of the manuscript that addresses the points raised during the review process.

We look forward to receiving your revised manuscript.

Kind regards,

Aziz ur Rahman Muhammad

Academic Editor

PLOS ONE

Journal Requirements:

2. Thank you for submitting the above manuscript to PLOS ONE. During our internal evaluation of the manuscript, we found significant text overlap between your submission and previous work in the [introduction, conclusion, etc.].

Please revise the manuscript to rephrase the duplicated text, cite your sources, and provide details as to how the current manuscript advances on previous work. Please note that further consideration is dependent on the submission of a manuscript that addresses these concerns about the overlap in text with published work.

[If the overlap is with the authors’ own works: Moreover, upon submission, authors must confirm that the manuscript, or any related manuscript, is not currently under consideration or accepted elsewhere. If related work has been submitted to PLOS ONE or elsewhere, authors must include a copy with the submitted article. Reviewers will be asked to comment on the overlap between related submissions (http://journals.plos.org/plosone/s/submission-guidelines#loc-related-manuscripts).]

We will carefully review your manuscript upon resubmission and further consideration of the manuscript is dependent on the text overlap being addressed in full. Please ensure that your revision is thorough as failure to address the concerns to our satisfaction may result in your submission not being considered further

Additional Editor Comments (if provided):

Dear Authors,

Although one of the reviewer has rejected the manuscript, however, i would like to invite you to review your manuscript according to suggestion of other reviewer.

Regards

Reviewers' comments:

Reviewer's Responses to Questions

**Comments to the Author**

1. Is the manuscript technically sound, and do the data support the conclusions?

Reviewer #1: No

Reviewer #2: Yes

Reviewer #3: Yes

Reviewer #4: Yes

2. Has the statistical analysis been performed appropriately and rigorously? 

Reviewer #1: No

Reviewer #2: Yes

Reviewer #3: Yes

Reviewer #4: I Don't Know

3. Have the authors made all data underlying the findings in their manuscript fully available?

Reviewer #1: Yes

Reviewer #2: Yes

Reviewer #3: Yes

Reviewer #4: Yes

4. Is the manuscript presented in an intelligible fashion and written in standard English?

Reviewer #1: Yes

Reviewer #2: Yes

Reviewer #3: Yes

Reviewer #4: Yes

5. Review Comments to the Author

Reviewer #1: In this study, authors used fistulated female camels (n = 3) to explore the particle-attached bacterial community for three types of plants after 6 and 12 hours of in situ incubation. For each plant, they used either acetone-extracted or non-extracted plant material. In addition to the bacterial community, this study explores the degradation of plant materials for up to 12 hours.

The primary concern in this study is the number of samples used to study the plant-attached bacterial community. The authors (lines 109–118) used only one bag for each category of sample (plant type, sample, incubation time) per animal. For each sample, one bag per animal is insufficient to capture the bacterial community. The authors did not run a permutational multivariate ANOVA (PERMANOVA) or other analysis to see if there were any statistical differences between bacterial communities for beta diversity analysis (lines 198–201).

Line 133-134: Several studies have shown that higher PCR cycle numbers result in chimaera generation and interfere with the analysis of bacterial community structure (1 & 2). This study used 35 PCR cycles to amplify the targeted region that has a higher chance of nonspecific amplification and errors. Generally, researchers are using up to 25 PCR cycles to amplify the targeted region in rumen bacterial communities.

Lines 76–78: Another aim of the study was to see the degradation of plant material, and this study only used two-time points (6 and 12 hours). These time points are not enough to see the degradation of the plant materials.

This manuscript is not technically sound in its current form. It lacks an adequate sample size, the number of replications, and degradation time points to be published.

References:

1- Suzuki, M. T. & Giovannoni, S. J. Bias caused by template annealing in the amplification of mixtures of 16S rRNA genes by PCR. Appl. Environ. Microbiol. 62, 625–630 (1996).

2-Gohl, D. M. et al. Systematic improvement of amplicon marker gene methods for increased accuracy in microbiome studies. Nat. Biotechnol. 34, 942–949 (2016).

Reviewer #2: i would like to thank the authors for the well written manuscript which provide good results to improve the usage of some shrubs and plants as alternative feed stuffs in animal nutrition in arid and semi arid regions which can improve animal performance

Reviewer #3: Comments to the Author

PONE-D-22-27641

The research topic of this manuscript in really interesting, since it focuses on changes in the bacterial community colonizing extracted and non-extracted tannin-rich plants in the rumen of dromedary camels. The manuscript is well articulated with minor revisions.

-Please add the P value in the results of the abstract and the results part.

-Please check that all abbreviations were provided in full names in their first mention and do not repeat.

- Line 50-51: In response to this sentence, the authors can say camel can utilize poor-quality fodder plants. Please authors can take a deeper look at this recent study it may be useful for the introduction and the discussion part as well.

Khattab, I.M., Abdel-Wahed, A.M., Anele, U.Y., Sallam, S.M. and El-Zaiat, H.M., 2021. Comparative digestibility and rumen fermentation of camels and sheep fed different forage sources. Animal Biotechnology, pp.1-10. doi: 10.1080/10495398.2021.1990939.

-Please, insert the method to calculate DM, CP and NDF disappearance.

-Table 1: please, check the chemical composition of the experimental fodder shrubs CP, total phenols, and total tannins. The CP is very high. Atriplex has low total phenols and condensed tannin content at 0.2g/kg DM according to ref 4 and khattab, 2007 (Khattab IM. Studies on halophytic forages as sheep fodder under arid and semi arid conditions in Egypt. Ph. D. Thesis. Alexandria University; 2007.(10.13140/RG.2.2.20956.21128)). Please authors can take a deeper look at this PhD study that was carried out in the same place as the study.

-Discussion needs to revise according to the chemical composition of fodder shrubs.

- Authors gave reasons for the low disappearance in acacia but did not give reasons for the high disappearance of Atriplex. The authors can benefit from this study, which shows the higher disappearance in the Atriplex due to the higher NPN content of the plant.

https://doi.org/10.1111/jpn.13577

Khattab, I.M. and Anele, U.Y., 2022. Dry matter intake, digestibility, nitrogen utilization and fermentation characteristics of sheep fed Atriplex hay‐based diet supplemented with discarded dates as a replacement for barley grain. Journal of Animal Physiology and Animal Nutrition, 106(2), pp.229-238.

Reviewer #4: The manuscript entitled “Changes in the bacterial community colonizing extracted and non-extracted tannin-rich plants in the rumen of dromedary camels.”

by Alaa Emara Rabee et al. is well written manuscript and will be of interest to the science community in the field. I have few minor comments for correction in the manuscript attached and will recommend the manuscript for publication in Plos One after the minor comments were answered properly.

6. PLOS authors have the option to publish the peer review history of their article (what does this mean?). If published, this will include your full peer review and any attached files.

Reviewer #1: No

Reviewer #2: No

Reviewer #3: **Yes: **Ibrahim M. khattab

Reviewer #4: No

---

## [Author Response · Author response to Decision Letter 0]

22 Jan 2023

Manuscript title: “Changes in the bacterial community colonizing extracted and non-extracted tannin-rich plants in the rumen of dromedary camels”

In the unclean or marked manuscript you could notice that colored comments using yellow.

Responses

Reviewer 1 

The primary concern in this study is the number of samples used to study the plant-attached bacterial community. The authors (lines 109–118) used only one bag for each category of sample (plant type, sample, incubation time) per animal. For each sample, one bag per animal is insufficient to capture the bacterial community.

>> Thank you for your comment, most of the studies use one bag per every type of sample to study the microbial community attached to the plants, and this is completely correct, especially we used three animals per group. However, using more than one bag is better, especially if the authors have enough funds. In our case, we used one bag and only two-time points due to the lack of fund.

The authors did not run a permutational multivariate ANOVA (PERMANOVA) or other analysis to see if there were any statistical differences between bacterial communities for beta diversity analysis (lines 198–201).

>> Thank you for your advice, we conducted PERMANOVA and modified the material and results sections.

Line 133-134: Several studies have shown that higher PCR cycle numbers result in chimaera generation and interfere with the analysis of bacterial community structure (1 & 2). This study used 35 PCR cycles to amplify the targeted region that has a higher chance of nonspecific amplification and errors. Generally, researchers are using up to 25 PCR cycles to amplify the targeted region in rumen bacterial communities.

>> Thank you so much for your comments, some Authors used these primers set for PCR amplification using 25 cycles and other used 35 cycles. However, an optimization study by Walters et al. (2015) used 35 cycles. Therefore, we used 35 cycle in our study.

Lines 76–78: Another aim of the study was to see the degradation of plant material, and this study only used two-time points (6 and 12 hours). These time points are not enough to see the degradation of the plant materials.

>> Thank you, we wanted to link the degradation with the changes in microbial communities, but due to the lack of the funds we used only two times. However, in the future work we will avoid this point.

Reviewer 2

Line 19: add and semi-arid before countries.

>>Thank you for your comments, modified

Line 31: you may add one sentence about the impact of the study on improving the animal performance.

>> added

Line 41: add alternative before solutions.

>> added

Line 51: add plant before species.

>> added

Line 58: add process after fermentation.

>> added

Line 82: change on to in.

>> Changed

Line 101: change female camels to be she-camels through all the manuscript.

>> Changed

Line 111: why the author select only two incubation times at 6 and 12 h …..For example why he did not increased the incubation period to reach 24 h or 48 hours.is there an explanation for that.

>> Thank you for your comment; we wanted to link the plant degradation with the changes in bacterial communities attached the plants. However, due to the lack of the fund we used only two incubation times. I hope we avoid this point in the future studies.

Line 174: add a legend for the table containing the full name of the abbreviations CP, NDF and ADF.

>> Thank you, we added it.

Line 190: same as the previous comment.

>> Thank you, we added it.

P value in table 2, 3, 4 and 5 to be centered.

>> Fixed

In the references part the doi name should be stable through all the references to be as follows https://doi.org/10.1007/s11250-011-9966-2.

>>Modified

Reviewer 3

-Please add the P value in the results of the abstract and the results part.

>> Thank you for your comments, added.

-Please check that all abbreviations were provided in full names in their first mention and do not repeat.

>> Checked and modified, thank you.

- Line 50-51: In response to this sentence, the authors can say camel can utilize poor-quality fodder plants. Please authors can take a deeper look at this recent study it may be useful for the introduction and the discussion part as well.

Khattab, I.M., Abdel-Wahed, A.M., Anele, U.Y., Sallam, S.M. and El-Zaiat, H.M., 2021. Comparative digestibility and rumen fermentation of camels and sheep fed different forage sources. Animal Biotechnology, pp.1-10. doi: 10.1080/10495398.2021.1990939.

>> Modified, and we used the reference.

-Please, insert the method to calculate DM, CP and NDF disappearance.

>> Clarified, thank you.

-Table 1: please, check the chemical composition of the experimental fodder shrubs CP, total phenols, and total tannins. The CP is very high. Atriplex has low total phenols and condensed tannin content at 0.2g/kg DM according to ref 4 and khattab, 2007 (Khattab IM. Studies on halophytic forages as sheep fodder under arid and semi arid conditions in Egypt. Ph. D. Thesis. Alexandria University; 2007.(10.13140/RG.2.2.20956.21128)). Please authors can take a deeper look at this PhD study that was carried out in the same place as the study.

>> Thank you for this comment, we recalculated the chemical composition based on the Reviewer advice, but we got the same results. The difference in the chemical composition in our study and khattab, (2007) might be attributed to using different plant part or collecting the plants in different season, we modified the discussion based on that and referred to the reference.

-Discussion needs to revise according to the chemical composition of fodder shrubs.

>> Thank you, please see the response above.

- Authors gave reasons for the low disappearance in acacia but did not give reasons for the high disappearance of Atriplex. The authors can benefit from this study, which shows the higher disappearance in the Atriplex due to the higher NPN content of the plant.

Khattab, I.M. and Anele, U.Y., 2022. Dry matter intake, digestibility, nitrogen utilization and fermentation characteristics of sheep fed Atriplex hay‐based diet supplemented with discarded dates as a replacement for barley grain. Journal of Animal Physiology and Animal Nutrition, 106(2), pp.229-238. https://doi.org/10.1111/jpn.13577

>> Thank you for this comments, we modified the discussion accordingly, and we used this reference.

Reviewer 4

The manuscript entitled “Changes in the bacterial community colonizing extracted and non-extracted tannin-rich plants in the rumen of dromedary camels.” by Alaa Emara Rabee et al. is well written manuscript and will be of interest to the science community in the field. I have few minor comments for correction in the manuscript attached and will recommend the manuscript for publication in Plos One after the minor comments were answered properly.

Thank you for your comments; we answered your comments as indicated in the attached file.

---

## [Decision Letter · Decision Letter 1]

27 Feb 2023

Changes in the bacterial community colonizing extracted and non-extracted tannin-rich plants in the rumen of dromedary camels

PONE-D-22-27641R1

Dear Dr. Rabee,

We’re pleased to inform you that your manuscript has been judged scientifically suitable for publication and will be formally accepted for publication once it meets all outstanding technical requirements.

Kind regards,

Aziz ur Rahman Muhammad

Academic Editor

PLOS ONE

Additional Editor Comments (optional):

Dear Authors

Authors have addressed all the comments/queries raised by reviewers. I am satisfied with manuscript now, please accept it in the present form

Reviewers' comments:

Reviewer's Responses to Questions

**Comments to the Author**

1. If the authors have adequately addressed your comments raised in a previous round of review and you feel that this manuscript is now acceptable for publication, you may indicate that here to bypass the “Comments to the Author” section, enter your conflict of interest statement in the “Confidential to Editor” section, and submit your "Accept" recommendation.

Reviewer #2: All comments have been addressed

Reviewer #3: All comments have been addressed

Reviewer #4: All comments have been addressed

2. Is the manuscript technically sound, and do the data support the conclusions?

Reviewer #2: Yes

Reviewer #3: Yes

Reviewer #4: Yes

3. Has the statistical analysis been performed appropriately and rigorously? 

Reviewer #2: Yes

Reviewer #3: Yes

Reviewer #4: I Don't Know

4. Have the authors made all data underlying the findings in their manuscript fully available?

Reviewer #2: Yes

Reviewer #3: Yes

Reviewer #4: Yes

5. Is the manuscript presented in an intelligible fashion and written in standard English?

Reviewer #2: Yes

Reviewer #3: Yes

Reviewer #4: Yes

6. Review Comments to the Author

Reviewer #2: thanks to the authors for that manuscript. i have no comments for this round if review. i recommend to accept the manuscript

Reviewer #3: I am satisfied with present version, the authors have been well responded.

no further comments .

Reviewer #4: I do not have additional comments. All my concerns has been addressed properly. Congratulation on publication in Plos One.

7. PLOS authors have the option to publish the peer review history of their article (what does this mean?). If published, this will include your full peer review and any attached files.

Reviewer #2: No

Reviewer #3: **Yes: **Ibrahim Mohamed khattab

Reviewer #4: No

---

## [Editor Report · Acceptance letter]

2 Mar 2023

PONE-D-22-27641R1 

Changes in the bacterial community colonizing extracted and non-extracted tannin-rich plants in the rumen of dromedary camels 

Dear Dr. Rabee:

I'm pleased to inform you that your manuscript has been deemed suitable for publication in PLOS ONE. Congratulations! Your manuscript is now with our production department. 

Kind regards, 

on behalf of

Dr. Aziz ur Rahman Muhammad 

Academic Editor

PLOS ONE